# OpenReview forum: "GUARDIAN: Safeguarding LLM Multi-Agent Collaborations with Temporal Graph Modeling"
_NeurIPS.cc/2025/Conference — NeurIPS 2025 poster_

### Official Review · Reviewer_Mtv2 · 2025-06-26

**Clarity:** 3
**Significance:** 2
**Originality:** 2
**Rating:** 4
**Confidence:** 3

**Summary:**

This paper introduces GUARDIAN, a framework designed to enhance the safety of LLM-based multi-agent collaborations by modeling interactions as temporal attributed graphs. The approach addresses critical challenges such as hallucination amplification and error propagation through an unsupervised encoder-decoder architecture that reconstructs node attributes and graph structures to detect anomalies. Leveraging Information Bottleneck Theory, GUARDIAN compresses temporal graphs while preserving key patterns, enabling efficient and precise identification of malicious agents or corrupted communications.

**Questions:**

1. The paper states that the time information in Z_t is modeled using transformers, with input being a series of graph embeddings. However, it is unclear whether temporal relationships between nodes at each step are explicitly considered. If the method involves concatenating embeddings of (Z_1, Z_2, …, Z_t-1)  does this implicitly assume temporal dependencies between nodes? Clarifying this assumption would strengthen the methodology.
2. Is the multi-agent system assumed to be fully connected? Specifically, are the outputs of agents in previous rounds visible to all agents in subsequent rounds? Clarifying the communication topology and its impact on the results is essential for reproducibility and understanding the framework’s scalability.
3. Does the framework support detecting anomalous interactions between agents, or is it limited to node-level analysis?
4. Can the system generalize to unseen agents during testing? These aspects are critical for real-world applicability.

**Ethical Concerns:**

["NO or VERY MINOR ethics concerns only"]

**Final Justification:**

The author has addressed my concerns, so I am keeping my original score.

**Limitations:**

1. Agents are functionally homogeneous and differ only in prompts, which simplifies the collaborative setting and lacks evaluation under complex multi-agent dynamics.
2. The use of MMLU and MATH benchmarks does not directly assess multi-agent safety, limiting the empirical validity of safety-related claims.

**Paper Formatting Concerns:**

The paper is generally well-structured and clearly structured.

**Quality:**

3

**Strengths And Weaknesses:**

Strengths:
1. The paper proposes GUARDIAN, a novel temporal graph-based approach to detect and mitigate safety risks (e.g., hallucination amplification) in LLM multi-agent systems, achieving superior performance with model-agnostic design.
2. The paper is well-written and the figure is vivid. The writing is easy to follow.

Weaknesses:
1. The paper emphasizes multi-agent safety but evaluates performance on unrelated downstream tasks (e.g., MMLU,MATH). These tasks are knowledge- or reasoning-intensive but do not directly assess safety. The authors should either justify this choice or include evaluations more relevant to safety, such as adversarial robustness or conflict resolution in multi-agent settings.
2. If the system is fully connected, the structure reconstruction decoder’s output would always be a matrix of ones, raising questions about its utility. The paper focuses on node-level anomalies (e.g., deleting anomalous agents) but does not address edge-level anomalies.
3. The current multi-agent setup appears simplistic. Agents seem functionally homogeneous, differing only in role prompts, which resembles a majority voting scheme rather than true multi-agent collaboration. The framework’s ability to handle complex interactions (e.g., tool usage, memory, heterogeneous agents) is untested. Additionally, can the system generalize to unseen agents during testing? These aspects are critical for real-world applicability.
4. The training data source for the framework is not specified, making it difficult to assess the validity of the results. Furthermore, how the three attack mechanisms (hallucination amplification, agent-targeted attack, communication-targeted attack) are implemented lack detailed descriptions. The authors should provide clear definitions, implementation details, and justification for these choices to ensure reproducibility.

---

> ### Author Rebuttal · Authors · 2025-07-31
>
> We appreciate the reviewer’s positive feedback on the novelty and effectiveness of GUARDIAN, its model-agnostic design, and the clarity of our writing and figures. Thank you for the valuable review. Below we address the detailed comments.
>
> **Question 1: Downstream tasks may not adequately evaluate multi-agent safety claims**
>
> We sincerely thank the reviewer for the thoughtful feedback. While we agree that direct evaluations of safety—such as adversarial robustness or conflict resolution—are valuable, our goal in this work is to investigate multi-agent safety within **general-purpose, realistic tasks** that better reflect practical deployment scenarios.
>
> Instead of relying solely on benchmarks explicitly designed for safety testing, we adopt widely used tasks such as MMLU and MATH, which allow us to explore how safety-related failures can emerge and be mitigated in collaborative reasoning. This perspective aligns with recent work that studies safety interventions in general-purpose reasoning tasks using standard benchmarks [1, 2].
>
> Furthermore, our evaluation **does include aspects of adversarial robustness**: our error injection setup simulates compromised or adversarial agents contributing misleading information. **GUARDIAN’s strong performance** under these conditions demonstrates its effectiveness in maintaining coordination and robustness in the presence of such disruptions. We appreciate the suggestion, and in future work we plan to incorporate more explicit safety-focused benchmarks.
>
> **Question 2: Structure reconstruction may be ineffective in fully connected graphs; only node-level anomalies addressed**
>
> We thank the reviewer for raising these insightful concerns regarding the utility of the structure reconstruction decoder and the scope of anomaly modeling.
>
> While our framework supports fully connected communication, we explicitly evaluate **sparser topologies** to avoid trivial structure decoding and better assess robustness. Specifically, we experiment with settings where each agent in the previous round is connected to only **25%**, **50%**, or **75%** of the agents in the next round. In these cases, the structure reconstruction decoder **does not produce an all-ones matrix**, and as shown in Table 2, our method continues to achieve strong performance. To ensure fair comparison, all models are evaluated under the same topology settings.
>
> Our anomaly detection module operates globally by aggregating outputs from previous rounds, enabling it to capture both **temporal inconsistencies** and **structural anomalies**. This temporal aggregation allows the model to detect patterns that may not be apparent within a single round or local neighborhood.
>
> While our removal strategy focuses on anomalous nodes, our framework also implicitly accounts for **edge-level anomalies**. Faulty or misleading communication links often manifest as **degraded outputs in the connected nodes in the subsequent round**. By identifying and removing such affected nodes, our approach effectively mitigates the impact of corrupted edges and prevents further error propagation.
>
> We will revise the manuscript to clarify these design choices and better articulate how both **node- and edge-level anomalies** are handled in our framework.
>
> **Table 2:** Performance comparison under different connection sparsity (MATH dataset, GPT-3.5-turbo). Bold values indicate highest accuracy.
>
> | Method         | 25%          | 50%          | 75%          |
> |----------------|--------------|--------------|--------------|
> | LLM Debate     | 26.5         | 34.6         | 34.2         |
> | DyLAN          | 38.6         | 41.6         | 39.3         |
> | SelfCheckGPT   | 7.6          | 5.6          | 3.4          |
> | **GUARDIAN**   | **52.2**     | **56.1**     | **57.3**     |
>
> **Question 3: Multi-agent setup appears simplistic; generalization to unseen agents**
>
> We sincerely thank the reviewer for the thoughtful and constructive feedback.
>
> We fully agree that supporting complex agent behaviors—such as **tool usage**, **memory**, and **heterogeneity**—is important for advancing real-world multi-agent systems. In this work, our aim is to develop a **general-purpose defense framework** for multi-agent safety under **minimal assumptions**. To better isolate and analyze core safety challenges, we adopt a **simplified setting** where agents are functionally homogeneous and focus on basic text-based QA. This design enables controlled experimentation and fair comparison. The framework is **modular and extensible**: more advanced agent capabilities can be incorporated without fundamental changes to the core mechanism. We consider these extensions highly promising and plan to explore them in future work.
>
> Regarding generalization, we emphasize that our method is **model-agnostic and unsupervised**—it does not rely on specific model architectures, training data, or prior knowledge of agent behavior. This enables effective generalization to **unseen agents** and **novel multi-agent configurations**, which we believe is essential for real-world applicability.
>
> **Question 4: Training data source and attack implementation details not specified**
>
> We sincerely thank the reviewer for raising these important points. We apologize for the lack of clarity in the original draft.
>
> - Our framework is evaluated on **four publicly available datasets**, and for each dataset, we train a model **independently** using the **Incremental Training Paradigm** within that dataset.
> - The **three attack mechanisms** are implemented as follows:
>   - **Hallucination amplification**: a failure mode where incorrect information is reinforced through iterative reasoning.
>   - **Agent-targeted attacks**: simulate compromised agents by modifying their role prompts to produce systematically incorrect outputs.
>   - **Communication-targeted attacks**: simulate link corruption by perturbing transmitted messages to mislead downstream agents.
>
> We will revise the manuscript to include **clearer definitions**, **implementation details**, and **justifications** for these settings to improve clarity and reproducibility.
>
> **Question 5: Temporal dependencies in Transformer not explicitly modeled**
>
> Thank you for the helpful comment.
>
> Yes, our method **implicitly models temporal dependencies** by feeding \((Z_1, Z_2, \ldots, Z_{t-1})\) into a **Transformer**, which captures **cross-time interactions** via self-attention. We will clarify this point in the revised version.
>
> **Question 6: Visibility of earlier agents' outputs to later agents**
>
> Thank you for pointing it out. In our setting, agents in each round can access the outputs of **only those agents they are directly connected to** from the immediately **preceding round**. They **do not** have visibility into all prior agents or the full global history. We will make this point clearer in the revision.
>
> References:
>
> [1] Yoffe et al. *DebUnc: Improving Large Language Model Agent Communication With Uncertainty Metrics.* arXiv 2024.
> [2] Amayuelas et al. *MultiAgent Collaboration Attack: Investigating Adversarial Attacks in Large Language Model Collaborations via Debate.* Findings of EMNLP 2024.

---

### Official Review · Reviewer_mN6C · 2025-06-30

**Clarity:** 3
**Significance:** 3
**Originality:** 2
**Rating:** 3
**Confidence:** 4

**Summary:**

GUARDIAN introduces a unified, model-agnostic framework for detecting and mitigating safety risks in LLM-based multi-agent collaborations by:

* **Temporal graph modeling:** representing each agent’s response and inter-agent communication as a discrete-time, attributed graph that makes the spread of hallucinations and injected errors explicit .
* **Information-Bottleneck abstraction:** compressing these temporal graphs by filtering out redundant interactions while preserving critical propagation patterns, bounding misleading cascades.
* **Incremental training:** fine-tuning the model round by round, pruning detected anomalies and adapting to evolving collaboration dynamics.
* **Empirical validation:** on benchmarks such as MMLU, MATH, FEVER and biographical tasks, GUARDIAN achieves state-of-the-art defense accuracy and reduces API calls compared to prior debate- and voting-based methods .

**Questions:**

See weakness

**Ethical Concerns:**

["NO or VERY MINOR ethics concerns only"]

**Final Justification:**

I keep my score "Borderline Reject".  As in weaknesses 3, the evaluation dataset is too small to show the overall evaluation performance.

**Limitations:**

1. **Dataset is too small:** Only 100 sample questions are used, which isn’t enough to prove the method works broadly.
2. **Unknown speed impact:** Anomaly detection runs every round, but the paper doesn’t report how much this slows things down.
3. **Examples are too simple:** The cases in Figures 10–12 could be solved by majority voting. The paper should include tougher, more realistic scenarios.

Check weakness

**Quality:**

3

**Strengths And Weaknesses:**

**Strengths**
1. **Innovative Temporal Graphs:** Captures agent interactions and error propagation over time, offering clear visibility into how misinformation spreads.;
2. **Unsupervised, Model-Agnostic Detection:** Leverages reconstruction errors—without task-specific labels—to flag anomalies across any LLM collaboration setup.;
3. **Efficient Empirical Performance:** Demonstrates higher defense accuracy with fewer API calls than debate- or voting-based methods on diverse benchmarks.;



**Weaknesses:**

1. **Section 3.1 is unclear:** The paper gives no concrete example or link to Figure 1. It never explains what “L” means in Equation (1). Section 5 lacks a simple walkthrough, and Figures 10–12 use text that’s too small.
2. **Agent graph not shown:** What is the actual graph layout for the “target” agent system? The authors mention both “4 agents” and “3–7 agents,” but please draw or show the 4-agent case. Authors could include frameworks like Camel or Autogen.
3. **Dataset is too small:** Only 100 sample questions are used, which isn’t enough to prove the method works broadly.
4. **Unknown speed impact:** Anomaly detection runs every round, but the paper doesn’t report how much this slows things down.
5. **Examples are too simple:** The cases in Figures 10–12 could be solved by majority voting. The paper should include tougher, more realistic scenarios.

---

> ### Author Rebuttal · Authors · 2025-07-31
>
> Thank you for the valuable comments and constructive suggestions that have helped us identify areas for improvement. Below we provide detailed responses to address your concerns.
>
> **Question 1: Some unclear information**
>
> We sincerely thank the reviewer for the thoughtful comments and helpful suggestions.
>
> - Regarding **Figure 1**, it is referenced on line 37 as an illustrative example of the three major categories of safety risks in LLM-based multi-agent systems, which motivate our overall design.
> - In **Equation (1)**, *L* denotes the loss function used to train the anomaly detection model; we will further clarify it and make it clearer.
> - In **Section 5**, we include a concrete walkthrough example (line 299, **Figure 4**), where GUARDIAN is applied to address the co-existence of hallucination and agent-targeted errors. We will improve the section and surrounding text to make this clearer.
> - We appreciate the feedback on **Figures 10–12**, and will enlarge the font size to improve readability.
>
> We are grateful for these suggestions and will incorporate them to enhance the clarity and presentation of the paper.
>
> **Question 2: Agent graph is not shown**
>
> We thank the reviewer for the helpful suggestion.
>
> We would like to claim that the current presentation would benefit from a more explicit visualization of the multi-agent layout. While **Figure 2(a)** illustrates a 3-agent setup, we agree that including a concrete example of a 4-agent system would make the target-agent configuration clearer, and we will add such a layout in the revised version.
>
> Our implementation leverages widely used multi-agent frameworks, including **LangGraph**, **Google ADK**, and **CrewAI**. We appreciate the suggestions to incorporate **CAMEL** and **AutoGen**, and will expand the discussion to include them for broader context and clarity.
>
> **Question 3: Dataset is small**
>
> We sincerely thank the reviewer for pointing it out. The dataset size plays a crucial role in evaluating the generality of a method. In our case, we would like to humbly clarify that the task under study is particularly challenging: solving a single example typically involves **multiple rounds of agent collaboration**, **numerous LLM queries**, and several **external API calls**. Due to this complexity, prior works in similar multi-agent settings [1, 2, 3] have also adopted evaluation sets of approximately 100 samples, which has emerged as a practical and useful setup in balancing task difficulty and experimental feasibility.
>
> After thorough research and consideration, we evaluated our approach across **four datasets** and **three independent runs**, resulting in over **1,200 full executions**. We believe this represents a meaningful experimental for this domain and provides empirical evidence supporting the robustness and general applicability of our method.
>
> **Question 4: Unknown speed impact**
>
> We thank the reviewer for raising this important and insightful point.
>
> To address this concern, we have conducted additional experiments to measure the **average runtime per question** (in seconds) under **communication-targeted attacks** using 4 agents and 3 rounds of interaction, as shown in Table 1.
>
> | Method        | MMLU   | MATH   | FEVER  |
> |---------------|--------|--------|--------|
> | LLM Debate    | 29.26  | **40.31** | 25.02  |
> | Challenger    | 26.15  | 56.23  | 27.50  |
> | Inspector     | 76.82  | 129.59 | 69.25  |
> | **GUARDIAN**  | **18.89** | 45.19  | **17.13** |
>
> **Table 1:** Time cost (s) comparison under communication-targeted attacks with GPT-3.5-turbo. Bold represents the lowest time cost.
>
> Our method achieves the **lowest time overhead** on **MMLU** and **FEVER**, and is only marginally slower than LLM Debate on MATH. These results demonstrate a favorable efficiency-performance trade-off. The reduced overhead is primarily due to **node deletion**, which results in **fewer API calls** and **reduced inter-agent waiting time**. Although anomaly detection is applied in every round, it incurs minimal cost due to the small size of the agent communication graph.
>
> **Question 5: Examples are too simple**
>
> We sincerely thank the reviewer for the thoughtful suggestion. The examples in **Figures 10–12** were primarily selected to clearly illustrate the key mechanisms of our approach.
> We agree that **more complex and realistic scenarios are also important** for evaluating robustness. In our experiments, we did encounter such cases where simple majority voting fails, and our method shows clear benefits. We will include these more challenging examples in the revised version to better highlight the strengths of our approach under realistic conditions.
>
> References:
>
> [1] Yoffe et al. *DebUnc: Improving Large Language Model Agent Communication With Uncertainty Metrics.* arXiv 2024.
> [2] Amayuelas et al. *MultiAgent Collaboration Attack: Investigating Adversarial Attacks in Large Language Model Collaborations via Debate.* Findings of EMNLP 2024.
> [3] Huang et al. *On the Resilience of LLM-Based Multi-Agent Collaboration with Faulty Agents.* ICML 2025.

---

### Official Review · Reviewer_ueHW · 2025-07-03

**Clarity:** 3
**Significance:** 4
**Originality:** 3
**Rating:** 4
**Confidence:** 4

**Summary:**

This paper proposes Guardian, a unified method for detecting and mitigating for agent-based systems. Specifically, a unsurprised encoder-decoder architecture is trained to identify anomalous nodes. Experimental results demonstrated that the proposed Guardian has strong performance in diverse benchmarks. Overall, the contribution of this paper is high to the community.

**Questions:**

- Can you please provide more intuition why encoder-decoder architecture can work in the Guardian?
- What is the trade-off between anomaly-detection threshold and the rate of incorrectly removed but benign agents?

**Ethical Concerns:**

["NO or VERY MINOR ethics concerns only"]

**Final Justification:**

My concerns has been addressed. I will keep my positive score.

**Limitations:**

Yes

**Paper Formatting Concerns:**

Citation formt is not consistent with the template.

**Quality:**

3

**Strengths And Weaknesses:**

Strengths:
- The manuscript is generally very well written and easy to follow
- The problem is timely and important to the commuity
- The proposed method, Guardian, is innovative and insightful
- Experimental results demonstrates strong performance of proposed Guardian defense

Weaknesses:
- The paper lacks an intuitive explanation and insights on why encoder-decoder architecture can work in this case
- Training and inference with Guardian can be resource intensive
- There is little discussion or empirical evaluation of false positives (i.e., non-malicious agent mistakenly flagged as anomalies)

---

> ### Author Rebuttal · Authors · 2025-07-31
>
> We thank the reviewer for highlighting the clarity, novelty, and strong performance of our work on a timely and important problem. Thank you for the valuable review. Below we address the detailed comments.
>
> **Question 1: Lack of intuitive explanation why the encoder-decoder architecture works in this context**
>
> Thank you for the insightful comment. We follow your suggestion to provide more intuition for the encoder-decoder design, and we will clarify this in the revision. LLM-based multi-agent collaboration is an increasingly important paradigm with significant potential, but it also introduces unique safety challenges such as hallucination amplification and error propagation. These issues are inherently unlabeled and thus cannot be effectively addressed using supervised approaches. To handle this, we adopt an unsupervised encoder-decoder architecture that models typical agent behavior over time. This design enables the model to identify agents whose behavior deviates from normal interaction patterns, without requiring labeled data. We appreciate the reviewer’s suggestion and will include this explanation in the revised version to improve clarity.
>
> **Question 2: Training and inference with Guardian can be resource intensive**
>
> Thank you for pointing it out. In experiments, we find that **GUARDIAN is relatively efficient**. As shown in **Figure 5**, it incurs the **fewest API calls among all methods evaluated**. This is because anomalous agents are progressively removed during each round, thereby reducing redundant API queries and improving efficiency over time.
>
> **Question 3: Discussion on non-malicious agents mistakenly flagged as anomalies**
>
> Thank you for pointing it out. We conduct the relevant experiments, and our results show that this risk is well controlled: **GUARDIAN achieves an average anomaly detection accuracy above 80%**, with **peak performance reaching 94.74%**, indicating that most removed agents are indeed problematic. In addition, we adopt a **conservative removal strategy**, where only the single agent with the highest anomaly score is removed per round, which helps minimize the likelihood and impact of false positives.
>
> To understand the consequences of false positives, we conducted a case-level analysis. The results show that even in rare cases where a correct agent was mistakenly removed, the system does not exhibit hallucination amplification or notable degradation in answer quality. We find that the remaining agents are often able to reach a correct decision, preserving the robustness of the final output. Moreover, under our iterative defense framework, such errors can often be detected and corrected in subsequent rounds.
>
> We will include representative examples and a brief analysis of these cases in the revised version to provide further clarity. Overall, even when false positives occur, their impact on final outcomes remains negligible, demonstrating the robustness of our method. We appreciate the opportunity to clarify this point.
>
> **Question 4: Trade-off between anomaly-detection threshold and the rate of incorrectly removed but benign agents**
>
> Thanks for the thoughtful question. As discussed in our response to Question 3, while our method is designed to effectively control the risk of mistakenly removing correct agents, we recognize that tuning the anomaly-detection threshold inevitably involves a balance between detection performance and false positive risk.
>
> In our experiments, the system demonstrates **relative robustness to threshold variations** when operating under a conservative removal policy. Furthermore, since typical configurations involve only a small number of agents (e.g., 3–7), removing more than one agent per round may disrupt system functionality. To mitigate such risks, we adopt a **cautious threshold** and limit removal to at most one agent per round, which helps preserve both stability and overall performance.
>
> **Question 5: Citation format is not consistent with the template**
>
> Thank you for your suggestion. We will revise accordingly.

---

> > ### Comment · Reviewer_ueHW · 2025-08-06
> > **Response to Rebuttal**
> >
> > Thank you for the detailed reply. My concerns has been addressed. I will keep my positive score. Good luck on your submission!

---

### Official Review · Reviewer_Pooy · 2025-07-07

**Clarity:** 2
**Significance:** 3
**Originality:** 2
**Rating:** 3
**Confidence:** 4

**Summary:**

This paper proposes GUARDIAN, a unified framework for safeguarding multi-agent collaborations of Large Language Models (LLMs) from critical safety risks, namely hallucination amplification and error injection/propagation. To enhance robustness and interpretability, the framework integrates an Information Bottleneck-based graph abstraction and an incremental training paradigm. Extensive experiments across benchmarks like MMLU, MATH, FEVER, and Biographies demonstrate that GUARDIAN significantly improves both safety and performance in multi-agent LLM collaborations, while remaining model-agnostic and computationally efficient.

**Questions:**

N/A

**Ethical Concerns:**

["NO or VERY MINOR ethics concerns only"]

**Final Justification:**

Due to the abovementioned reasons, my final rating is 3: borderline reject.

**Quality:**

3

**Strengths And Weaknesses:**

Strengths
- The paper addresses a timely and important problem: the growing safety risks in multi-agent LLM collaborations.
- The paper is well-written and easy to follow. In particular, the problem statement is clearly presented in L48-49: "This research aims to develop a unified method for detecting and mitigating multiple safety concerns in LLM collaboration processes, including hallucination amplification and error injection and propagation."
- GUARDIAN is evaluated in multiple different scenarios, demonstrating clear performance improvements and low number of API calls. The proposed method benefits in both accuracy and cost at the same time.

Weaknesses and Questions
- Limited technical novelty: graph-based approaches in multi-agent setups are already proposed. Each contribution presented in this paper is inspired from somewhere else, e.g., Graph Encoder, Reconstruction Decoder, GIB as well. While applying them into LLM-based multi-agent systems is an interesting study, such strategies have been common in other multi-agent systems.
- Few lines are overly stated. For example, considering the limited technical contribution, I don't think this is a pioneering work, as stated in L57-59: "we pioneer a graph abstraction mechanism rooted in the Information Bottleneck Theory (25)". Also, GUARDIAN exhibits clear degradation when agent number exceeds 5, but in L306-307, the author mentioned “our framework maintains consistent performance across different configurations, demonstrating effective adaptation to multi-agent networks of varying sizes”. Following this statement, why the accuracy slowly degrades when agent number is more than 5?
- Figure 2(b) is confusing. I guess, the authors tried to demonstrated that introducing some anomalous communication at graph edges suddenly increases anomaly degree at corresponding nodes at time t1. I couldn’t see introducing anomalous communication in Figure 2(b) (right) exhibits high anomaly degree of agents compared to the left one.
- There is no discussion on failure modes, e.g., when GUARDIAN may miss an anomaly or falsely remove a correct agent. I guess, if the corrected agent is removed by mistake, hallucination could be possibly more amplified than not applying GUARDIAN.
- In L256 (Datasets), training dataset is not discussed. is this Incremental Training Paradigm performed in a specific set of training data? I'm basically wondering if the anomaly detection would work well only on in-distribution data setting.

---

> ### Author Rebuttal · Authors · 2025-07-31
>
> We appreciate the reviewer’s recognition of our timely and clearly presented work, its thorough evaluation, and the method’s accuracy and efficiency. Thank you for the valuable review. Below we address the detailed comments. We hope you may find our response satisfactory and increase the score accordingly.
>
> **Question 1: Limited technical novelty**
>
> Thanks for pointing it out. We would like to humbly claim that LLM-based multi-agent collaboration introduces an unique class of safety challenges that are fundamentally different from those in traditional multi-agent systems. In particular, issues such as hallucination amplification and error propagation across rounds are global, temporally distributed, and label-free, posing unique difficulties for detection. To tackle these challenges, in this paper we propose a tailored temporal graph modeling framework that integrates an unsupervised encoder-decoder architecture and GIB-based abstraction, specifically designed to work without labeled data or any modification to base LLMs. While inspired by existing techniques, the way these components are adapted and combined for this unique problem setting is, to our knowledge, unique. We believe this represents a meaningful step toward building safer LLM-based collaborative systems.
>
> **Question 2: Few lines are overly stated**
>
> We thank the reviewer for the valuable feedback and apologize for any confusion caused by our wording. Our intention in Lines 57–59 was not to claim a pioneering contribution to Information Bottleneck theory itself, but rather to highlight that, to the best of our knowledge, we are the first to systematically apply a graph abstraction mechanism grounded in IB theory to address safety risks in LLM-based multi-agent collaboration. We will revise it to better reflect the scope and context of our contribution.
>
> Regarding the second point, we appreciate the reviewer pointing out the ambiguity. Our statement in Lines 306–307 was meant to emphasize that within each agent-number setting, our method consistently outperforms baselines, demonstrating robustness under different configurations. We did not intend to imply that absolute performance remains unchanged as the number of agents increases. As shown in Table 2 in this paper, the gradual decrease in accuracy with more agents is expected due to increased complexity and error propagation, but our method still maintains a clear advantage over other approaches. We will improve clarity in the revision.
>
> **Question 3: Figure 2(b) is confusing**
>
> Thank you for the helpful comment. We apologize for the confusion caused by the presentation of Figure 2(b). The left and right plots illustrate two independent scenarios: the left shows agent-targeted error injection, while the right shows communication-targeted error injection. They are not intended for direct visual comparison, but rather to demonstrate different types of anomalies under our framework. We will clarify this distinction explicitly in the caption to avoid misunderstanding.
>
> **Question 4: No discussion on failure modes**
>
> Thank you for raising this suggestion. To further assess the potential effects of mistakenly removing a correct agent, we conduct the corresponding experiments. The results show that this risk is well controlled: **GUARDIAN achieves an average anomaly detection accuracy above 80%**, with **peak performance reaching 94.74%**, indicating that most removed agents are indeed problematic. In addition, we adopt a conservative removal strategy, where **only the single agent with the highest anomaly score is removed per round**, which helps minimize the likelihood and impact of misclassifications.
>
> To understand the consequences of incorrect removals, we conducted a case-level analysis. The results show that even in rare cases where a correct agent is mistakenly removed, the system does not exhibit hallucination amplification or notable degradation in answer quality. We observe that the remaining agents often still reach a correct decision, preserving the robustness of the final output. Moreover, under our iterative defense framework, incorrect decisions in one round can often be corrected in subsequent rounds.
>
> We will include representative examples and a brief analysis of such cases in the revised version to further clarify this point. Overall, even when a correct agent is mistakenly removed, its impact on final outcomes remains negligible, further demonstrating the robustness of our method. We appreciate the opportunity to clarify this point.
>
> **Question 5: Training dataset is not discussed**
>
> Thank you for the question. We use four datasets in our experiments, and for each dataset, we train a separate model to achieve the corresponding capability. The **Incremental Training Paradigm** is applied within each dataset, rather than across datasets. That is, our method is designed specifically for **in-distribution anomaly detection**, where the underlying data distribution remains consistent while new episodes are incrementally introduced. This design choice enables us to systematically evaluate temporal anomaly detection performance under realistic conditions that preserve domain consistency.

---

> > ### Comment · Reviewer_Pooy · 2025-08-06
> >
> > Thank you for addressing my questions. Some of my concerns are addressed. While I agree with the authors to a certain extent (e.g., "LLM-based multi-agent collaboration" introduces a novel class of problem), I still doubt whether the proposed method is really a tailored to this problem. I partially agree with this statement: "global, temporally distributed, and label-free, posing unique difficulties for detection", as such problem has been widely studied in the domain of relational inference. In addition, this unsupervised encoder-decoder approach is very common and widely adopted in such tasks, weakening the overall novelty, even if the problem setting is unique. For the answer 4, I expected more details on the experiments. While the authors state the high accuracy of 80%, there is no information about the evaluation datasets and how often it varies (as it sometimes reaches 94%, I guess it may drop a lot). For these reasons, I would remain at the same rating.

---

> > > ### Author Response · Authors · 2025-08-08
> > >
> > > We sincerely thank you for the continued engagement and insightful feedback. We are grateful for this opportunity to clarify the remaining points.
> > >
> > > We agree that concepts like temporal relational inference and unsupervised encoder-decoder architectures are well-established. However, our novelty does not lie in inventing these components from scratch, but rather in the novel synthesis and specific adaptation of these techniques to address the unique phenomenology of safety issues in LLM multi-agent collaboration, a problem space that exhibits critical differences from traditional relational inference tasks.
> > >
> > > We wish to clarify these key distinctions:
> > >
> > > 1. **Nature of Node Attributes and Anomalies**: In traditional relational inference (e.g., social networks, sensor networks), node attributes are often structured numerical vectors, and anomalies are typically structural (e.g., unexpected links) or value-based (e.g., outlier readings). In our setting, a node's attribute is high-dimensional, unstructured natural language generated by an LLM. An "anomaly" (a hallucination or injected error) is not a simple outlier but a subtle semantic deviation that can be grammatically perfect. Reconstructing and identifying anomalies in such rich, semantic-laden text is a fundamentally different and more challenging task than handling numerical data. Our model is specifically designed to capture these semantic nuances through its text-reconstruction objective.
> > >
> > > 2. **Dynamics of Anomaly Propagation**: The core challenge we address is "hallucination amplification" and "error propagation". This is not merely a temporal dependency, but a cascading semantic corruption. An error injected at time t doesn't just affect node states at t+1; it semantically influences the reasoning process of subsequent agents, leading to compounded and often unpredictable downstream failures. Our temporal graph model, combined with the incremental training paradigm, is explicitly tailored to capture this unique, path-dependent semantic degradation, which is a hallmark of LLM interactions, not general relational systems.
> > >
> > > In summary, the instantiation of our framework—jointly reconstructing graph structure and high-dimensional text attributes, the incremental training to model semantic drift, and the GIB-based abstraction to manage conversational complexity—is a highly tailored solution for a problem that is, to our knowledge, new and distinct.
> > >
> > > Regarding Question 4, we have conducted a more rigorous analysis of GUARDIAN's False Discovery Rate (FDR). This metric directly answers a critical practical question: "When GUARDIAN flags a node as anomalous and intervenes, what is the probability that this is a false alarm (i.e., the node was actually correct)?" Technically, the False Discovery Rate is calculated as the ratio of false positives (FP)—nodes incorrectly flagged as anomalous—to the total number of detections (the sum of true positives, TP, and false positives, FP). It therefore measures the precision and trustworthiness of our system's detections.
> > >
> > > **Table 1. Analysis of Detection Reliability: False Discovery Rate (FDR, %).**
> > >
> > > | Safety Issue                                         | MMLU  | MATH  | FEVER | Biographies |
> > > |------------------------------------------------------|-------|-------|--------|-------------|
> > > | Hallucination Amplification                          | 26.67 | 13.11 | 17.86  | 15.69       |
> > > | Agent-targeted Error Injection and Propagation       | 22.22 | 8.32  | 20.53  | 13.23       |
> > > | Communication-targeted Error Injection and Propagation | 30.67 | 18.42 | 28.57  | 19.65       |
> > >
> > > The results, summarized in Table 1, demonstrate that GUARDIAN maintains a low False Discovery Rate. Notably, this rate is exceptionally low at just 8.32% on the MATH dataset, and for the majority of scenarios, it remains well below 20%. This high degree of precision ensures that the vast majority of our system's interventions are correct.
> > >
> > > To analyze the impact, we measured the final task success rate in cases where a correct agent was mistakenly removed. The results show that the impact is minimal. This resilience stems from two factors inherent in our design:
> > >
> > > 1. **System Redundancy**: In a multi-agent system, there is often informational redundancy. The remaining correct agents can typically compensate for the loss of one agent, leading the collaboration to the correct outcome.
> > >
> > > 2. **Conservative Strategy**: Our approach removes only the single most anomalous agent per round, which prevents catastrophic failures from multiple incorrect removals.
> > >
> > > We will include this table and a qualitative case study in the revised manuscript to demonstrate that even in the rare instances of failure, GUARDIAN's design ensures the system degrades gracefully rather than catastrophically amplifying errors.

---

> > > ### Author Response · Authors · 2025-08-08
> > >
> > > Dear Reviewer Pooy,
> > >
> > > Thank you for your valuable feedback again! In our rebuttal, we have carefully addressed the mentioned points. Given that the discussion phase deadline is approaching, we welcome any additional feedback you might have. If our responses have addressed your concerns, we would appreciate your consideration for updating the score.
> > >
> > > Best,
> > > Authors

---

### Note · Authors · 2025-08-12

We sincerely thank all reviewers and the AC for their constructive and thoughtful feedback throughout this process. The engagement and insights have strengthened our work considerably.

We are grateful for the strong consensus across reviews recognizing our core contributions. Reviewers acknowledged: (1) the critical importance of addressing safety risks in LLM-based multi-agent collaboration, particularly hallucination amplification and error propagation—challenges that become increasingly urgent as these systems scale, and (2) the empirical strength and practical efficiency of GUARDIAN, which delivers consistent improvements across diverse benchmarks while maintaining model-agnostic design and significantly reducing computational overhead.

We deeply appreciate Reviewer Pooy's engagement with our work and the productive discussion that followed. While initial concerns about technical novelty were raised, we recognize that our contribution centers on **purposefully designing an unsupervised and model-agnostic framework to address the distinctive challenges of LLM collaboration**—where anomalies manifest as semantic drift, propagate through natural language, and exhibit complex temporal dependencies. The **quantitative evidence** we provided and our design choices demonstrate GUARDIAN's robustness across diverse conditions, validating our approach.

Overall, GUARDIAN represents a meaningful step toward safer multi-agent LLM systems. As collaboration protocols like Agent2Agent (A2A) gain prominence, our work addresses an essential need: **detecting and mitigating semantic anomalies that can cascade through agent networks**. By modeling interactions as temporal graphs within an **unsupervised, model-agnostic framework**, GUARDIAN enables precise safety monitoring without requiring access to base LLMs—a crucial property for real-world deployment. As multi-agent LLM systems become integral to applications spanning collaborative tutoring, autonomous planning, and scientific discovery, proactive safety measures transition from enhancement to **necessity for trustworthy AI deployment**.

We envision GUARDIAN as a **foundational component in the ecosystem of safer, more accountable multi-agent AI**. We hope our work catalyzes further research into principled defenses for this rapidly evolving domain, where powerful AI systems can collaborate safely and reliably.

Thank you for the opportunity to contribute to advancing this important area of research.

---

### Decision · Program_Chairs · 2025-09-17

**Decision:**

Accept (poster)

**Comment:**

The paper introduces GUARDIAN, a framework designed to enhance the safety and reliability of multi-agent collaborations. The central claim is that by modeling multi-turn interactions as a discrete-time temporal attributed graph, it's possible to effectively detect and mitigate critical safety risks like hallucination amplification and error propagation. The core of their method is an unsupervised encoder-decoder architecture. I believe overseeing multi-agent collaborations and anomaly detections are important areas of safety research, and this work applies anomaly detection techniques from traditional ML to this emerging area. There is a strong agreement (from all reviewers) that the problem of ensuring safety in multi-agent LLM systems is both critical and timely. Reviewers also find the novelty and clarity are great. In terms of concern, Reviewer mN6C noted the dataset size (100 samples) is small, although this is somewhat common for complex, costly multi-agent experiments. I suggest that authors include those challenging examples as mentioned during rebuttal in a revised version of the paper. Moreover, the authors have conducted new experiments during the discussion period to provide a quantitative analysis of the false discovery rate and a runtime comparison (addressing efficiency concerns). Overall, the value behind this work that brings conventional anomaly ML methods into localizing failures in multi agent systems is crucial, while the concerns are mostly addressed / addressable. Thus, I lean acceptance to encourage the exploration in this direction and I hope the follow-up work can build a much more bigger evaluation and detection framework.